# Testing for Dihydropyrimidine Dehydrogenase Deficiency to Individualize 5-Fluorouracil Therapy

**DOI:** 10.3390/cancers14133207

**Published:** 2022-06-30

**Authors:** Robert B. Diasio, Steven M. Offer

**Affiliations:** 1Department of Molecular Pharmacology and Experimental Therapeutics, Mayo Clinic, Rochester, MN 55902, USA; diasio.robert@mayo.edu; 2Mayo Clinic College of Medicine and Science, Mayo Clinic, Rochester, MN 55902, USA

**Keywords:** pharmacogenetics, precision medicine, fluorouracil, chemotherapy, dihydropyrimidine dehydrogenase, adverse events

## Abstract

**Simple Summary:**

5-Fluorouracil (5-FU) is a chemotherapy drug that is commonly used to treat multiple cancers. Many people who are treated with 5-FU experience severe toxicity to the drug, and in severe cases, patients can die. This review discusses current methods for identifying people who are at high risk for severe side effects to 5-FU therapy.

**Abstract:**

Severe adverse events (toxicity) related to the use of the commonly used chemotherapeutic drug 5-fluorouracil (5-FU) affect one in three patients and are the primary reason cited for premature discontinuation of therapy. Deficiency of the 5-FU catabolic enzyme dihydropyrimidine dehydrogenase (DPD, encoded by *DPYD*) has been recognized for the past 3 decades as a pharmacogenetic syndrome associated with high risk of 5-FU toxicity. An appreciable fraction of patients with DPD deficiency that receive 5-FU-based chemotherapy die as a result of toxicity. In this manuscript, we review recent progress in identifying actionable markers of DPD deficiency and the current status of integrating those markers into the clinical decision-making process. The limitations of currently available tests, as well as the regulatory status of pre-therapeutic *DPYD* testing, are also discussed.

## 1. Introduction

The fluoropyrimidine analog 5-fluorouracil (5-FU) was introduced as an anti-cancer agent in the late 1950s and remains one of the most widely prescribed chemotherapeutics, with an estimated 2 million people worldwide receiving 5-FU or one of its prodrug forms (e.g., capecitabine) each year [1]. 5-FU is used to treat many types of cancers, most predominantly colorectal cancer, where it is used as a component of first-line adjuvant therapy and for advanced disease [2]. In addition, 5-FU continues to be used to treat breast and pancreatic cancers [3,4], among others. Despite being well-tolerated in general, therapy-related toxicity remains a high concern with 5-FU use. The prevalence of severe (clinical grade 3 or greater) toxicity varies by treatment regimen. Using data from a large prospective cooperative group clinical trial (Alliance N0147), investigators estimated that approximately one in three patients that received current-generation multi-drug regimens for the adjuvant treatment of colon cancer experienced grade 3 or higher toxicities that are typically associated with 5-FU use [5]. Similarly high rates of toxicity have been noted in other clinical trials utilizing 5-FU-based treatments [6,7,8]. However, it is noted that the co-administration of additional therapeutics in modern therapeutic approaches makes it difficult to pinpoint the exact number of toxicities that are specifically caused by 5-FU and not related to concomitant drugs or interactions between the components of multi-drug therapy [9].

Genetic factors are known to contribute to the risk of developing severe toxicity to 5-FU, with those related to decreased function of the enzyme dihydropyrimidine dehydrogenase (DPD) emerging as a critical determinant of toxicity risk. DPD is the initial and rate-defining step of the uracil catabolism pathway, which also converts 5-FU to inactive metabolites (Figure 1) [10]. In the 1980s, it was recognized that patients who had experienced toxicity to 5-FU tended to have elevated levels of uracil in the blood and urine [11,12], suggesting that hepatic DPD deficiency could be an underlying cause of 5-FU toxicity. The first case of DPD deficiency was confirmed by measuring DPD enzyme function in peripheral blood mononuclear cells (PBMCs), and genetic inheritance was confirmed by expanded pedigree analyses [10,12]. Subsequent studies within this family identified two deleterious variants in the gene encoding DPD (*DPYD*) that segregated independently and demonstrated an autosomal codominant pattern of inheritance [13]. The central role of DPD in determining 5-FU exposure and toxicity risk is further exemplified by the drug–drug interactions noted between 5-FU and antiviral uracil nucleoside analogs. The antiviral drug Sorivudine (1-beta-D-arabinofuranosyl-E-5-[2-bromovinyl] uracil) was lethal in patients treated with 5-FU [14,15,16]. The drug was later shown to inhibit hepatic DPD, leading to prolonged 5-FU exposure and increased anabolism of 5-FU to cytotoxic metabolites [14,17,18].

## 2. *DPYD* Variants Associated with 5-FU Toxicity

Several *DPYD* variants have been studied in clinical studies of 5-FU toxicity and in pre-clinical models of DPD function or 5-FU metabolism. Four variants have reproducibly shown significant association with elevated risk of severe toxicity to 5-FU (Table 1): c.1905+1G>A (DPYD*2A, IVS14+1G>A, rs3918290), c.1679T>G (DPYD*13, p.I560S, rs55886062), c.2846A>T (p.D949V, rs67376798), c.1129-5923C>G(rs75017182) [19,20]. These four variants demonstrate the wide variability of impacts that alleles can have on DPD enzyme activity and toxicity risk. Overall, carriers of these four risk alleles are estimated to be 1.6–4.4 times more likely to experience severe adverse events [19] and >25% more likely to experience lethal toxicity [21] to 5-FU compared to non-carriers.

The most studied *DPYD* variant, c.1905+1G>A causes obligate in-frame skipping of *DPYD* exon 14 [22,23,24], resulting in a catalytically inactive form of the protein [25,26]. Heterozygous carriers of c.1905+1G>A exhibit ~50% reduced DPD activity as measured ex vivo in peripheral blood mononuclear cells (PBMCs) [13,27,28] and display prolonged exposure to 5-FU and active metabolites [29].

The c.2846A>T allele was originally identified in a DPD-deficient family [30,31] and was later shown to be associated with severe 5-FU toxicity [19,32]. Direct in vitro study of this variant demonstrates that the translated protein retains partial DPD activity [25].

As the rarest of these four variants, c.1679T>G was also found to co-transmit with DPD deficiency within a pedigree surrounding a patient who experienced severe 5-FU toxicity [13]. Functional studies demonstrate that the DPD protein translated from c.1679T>G transcripts retains a low level of residual DPD activity [25]. While carriers of this variant are more likely to experience severe 5-FU toxicity, the rarity precludes conclusive clinical analyses [19], and the variant is the only one of these four that is not currently assigned a “strong” level of evidence for 5-FU toxicity association by the Clinical Pharmacogenetics Implementation Consortium (CPIC) [20].

As a less severely deleterious variant, c.1129-5923C>G creates a novel non-obligate splice donor site within intron 10 that leads to partial alternative splicing to include an additional out-of-frame exon [33,34,35]. This variant was originally identified as a collection of alleles that was termed “HapB3” [36]; later studies demonstrated that HapB3 tagged the deep-intronic splice-site variant rs75017182 [34], which was later shown to be causal [33]. A synonymous coding region variant c.1236G>A (p.E412E, rs56038477) is in near-perfect linkage disequilibrium with rs75017182 and is often used as a genotyping proxy [7]. While the impact on DPD function appears milder than other risk variants at the functional level, current evidence suggests that the contributions or rs75017182 to toxicity risk could vary by population (e.g., compare [7] and [37]), which might be due to differences in treatment and/or variable co-transmission of other alleles that potentially exert mild-to-moderate effects on DPD function [38].

While these four variants have been studied in depth, they are unlikely to be the only variants associated with risk. By measuring DPD activity ex vivo using PBMCs collected from a population of volunteer subjects, *DPYD*-c.557A>G (p.Y186C, rs115232898) was identified in individuals with self-declared African American race/ancestry [28] and has since been recognized as a risk variant (Table 1). Carriers of c.557A>G had significantly lower PBMC DPD activity compared to non-carriers [28]. The variant was later found in patients that suffered severe, and in one case lethal, toxicity to 5-FU [39,40,41]. In vitro characterization of p.Y186C confirmed that the variant was deleterious to function [39], and the variant is directly mentioned as a risk variant for 5-FU toxicity in the current CPIC Guideline for Fluoropyrimidines and *DPYD* [20]. Additional studies that were conducted in individuals of African ancestry identified multiple additional risk variants using sequencing coupled with in vitro functional analyses [42], suggesting that the contribution of previously unrecognized risk variants is likely higher in under-studied populations.

Most clinical studies and, by extension, meta-analyses have been conducted exclusively in Europe or in individuals of European ancestry (e.g., self-declared “white” individuals) [19,21,43,44]. The studies within African American populations demonstrated that the four well-studied risk variants discussed earlier are all but absent from ancestral African haplotypes [28,42]; additional studies of large publicly available sequence repositories strongly suggest that those variants are highly enriched in European/white ancestral haplotypes and are likely of limited utility as biomarkers in other populations [45].

Case reports and population-agnostic functional studies that utilize cellular or in vitro models of DPD deficiency have been instrumental in identifying additional candidate 5-FU risk variants within *DPYD*. Neonatal screening programs based in the Netherlands have identified numerous cases of pyrimidine imbalance that were linked to *DPYD* variants in both the domestic Netherlands and international populations [46,47,48,49]. Large-scale sequencing efforts have also identified hundreds of additional nonsynonymous variants of unknown significance in *DPYD*. The analysis of carriers and the use of patient-agnostic approaches to characterize these variants have greatly improved our understanding of the repertoire of DPD-deficiency-associated alleles [25,45,50,51,52].

## 3. Identifying DPD Deficiency

The clinical data linking DPD deficiency to 5-FU toxicity, as well as the well-studied metabolism pathway DPD as the major 5-FU catabolic enzyme, have made identifying patients with DPD deficiency a clinical priority for potential therapeutic dose adjustment. Varied methods of assessing DPD deficiency have been developed, each with potential advantages and disadvantages. These tests include genetic tests of varying coverage, the measurement of blood metabolites as an indicator of DPD function, measurement of DPD function directly in PBMCs as a proxy for liver function, and others. The following sections will outline these varied approaches and review the literature relevant to their use. Because of the rapid evolution occurring in the field of DPD testing, specific companies and products will not be named.

## 4. Genetic Tests to Identify DPD Deficiency

Genotype-based approaches to identify DPD deficiency are becoming more common and offer potential advantages over phenotypic tests, including high diagnostic accuracy with results that are not influenced by environmental factors or methodological differences in sample handling and processing [53]. As such, genetic tests have seen more wide-spread availability and the development of evidence-driven recommendations for dose adjustment based on genotype [20,54,55].

### 4.1. Targeted Genotyping for Specific DPYD Variants

Most genotypic tests for DPD deficiency use targeted assays to identify the allele status for individual preselected single nucleotide variations (SNVs). Some commercially offered tests only provide the genotype for a single *DPYD* variant, most commonly c.1905+1G>A, and do not genotype for any other risk variants. Therefore, it is important for users to understand the limitations of incomplete genotyping, especially since the more common causal alleles in Europeans (i.e., c.1129-5923C>G and c.2846A>T) might not be genotyped by a targeted test.

Within most European populations, targeted tests for the four well-studied variants discussed above will likely identify most carriers for variants associated with DPD deficiency. While individual risk variants outside of the four commonly studied variants are individually rare, when considered collectively, these variants are likely carried by a measurable fraction of the European population [45]. Furthermore, the recent discovery of multi-marker contributors to DPD activity indicate that targeted genotype panels may not be as comprehensive as previously believed and that expanded panels may be necessary to more accurately predict risk [38].

Unfortunately, targeted genetic tests likely have extremely limited utility in individuals with non-European ancestry, where the four well-studied risk variants are exceedingly uncommon and other variants predominate. As new information has been gained in the field, some testing laboratories are introducing expanded tests to keep pace with developments. For example, the c.557A>G variant and additional rare deleterious *DPYD* variants are now included on some targeted genotyping panels offered by a small number of testing laboratories. Given the discrepancies between test offerings from various laboratories, it is critical that those ordering genetics tests be aware of the variants that are included in a given test.

### 4.2. Sequence-Based Testing for DPD Deficiency and Interpretation of Novel Alleles

Targeted genotyping can be advantageous for cost and turnaround considerations; however, the tests do not provide information outside of the specific SNVs being tested. As an alternative approach, sequence-based genotype assessment is starting to be offered by some testing laboratories. Sequencing has the potential to identify deleterious variants that would be missed by variant-specific genotype methods [56], making it an appealing choice for patients with non-European ancestry who may carry other risk alleles that are not commonly included on targeted genetic tests.

While sequence-based genotyping has the potential to overcome some of the limitations with SNV-specific assays, the contributions to 5-FU toxicity risk for the variants that are detected may not always be interpretable. Treatment guidance may be available for carriers of some alleles [20]; however, if an identified variant has not been previously characterized and reported, there is no information on which to base treatment decisions. Prediction tools aimed at classifying unknown variants using large databases of generalizable information from other variants have long attempted to fill this gap with varying degrees of success. Generalized prediction tools, such as SIFT [57] and Polyphen [58], were developed by training models to assess generalized protein features for contribution to known genetic diseases. In pharmacogenetic conditions such as DPD deficiency, it is unclear if the same underlying principles of protein function apply since the consequences may not manifest until after treatment with a drug. As such, variants that do not cause an overt disease state in the absence of a compound’s use can still be pharmacologically relevant. Attempts to apply generalized protein prediction tools to pharmacogenomics have confirmed their low performance at distinguishing deleterious from benign pharmacovariants [45,59]

A new generation of prediction tools seeks to fill that gap. With respect to *DPYD* variants, we developed a gene-specific variant classifier that was developed using features intrinsic to *DPYD* and 5-FU metabolism and trained using a robust in vitro measure of SNV impacts on DPD enzyme activity [45]. Using extensive cross validation and independent variant sets, we were able to assess the accuracy of DPYD-Varifier at predicting which *DPYD* variants were deleterious. A comparison of this new tool with existing general classifiers demonstrated that the gene-specific tool was more accurate than conventional general tools and correctly classified all well-studied variants and most novel ones [45]. Additionally, Zhou et al. recently incorporated published in vitro functional data for missense *DPYD* and *TPMT* variants using an ensemble learning approach and confirmed that gene-specific variant classifiers have the potential to dramatically improve prediction accuracy for *DPYD* variants of unknown significance [60]. Companion analyses with additional class-specific tools such as MMSplice [61] and RegSNPs-intron [62] have the potential to further improve classification of *DPYD* variants identified through sequencing.

## 5. Phenotypic Methods to Identify DPD Deficiency

Phenotypic approaches for measuring DPD activity were developed as research tools to identify DPD deficiency and subsequently characterize the genetics of the condition in pedigrees linked to individuals with severe 5-FU toxicity (e.g., [12,30,31,46,63,64,65]). These approaches estimate the ability of DPD to catabolize 5-FU in vivo and have the potential to identify individuals with DPD deficiencies due to factors outside of known causal alleles detected by genetic tests. While phenotypic tests have been instrumental as research tools, they have not been as widely accepted in clinical decision making as genetic tests. This may be in part due to the high degree of variability noted within and between phenotypic DPD tests [28,66], particularly when specimens are collected and analyzed at more than one site [38,67]. In addition, while clear correlations with clinical 5-FU toxicity have been established at the individual level for genetic risk factors, the same level of evidence for toxicity association has not been demonstrated for phenotypic tests. Regardless, multiple attempts at establishing non-genetic tests for 5-FU toxicity risk have been made with varying degrees of success.

### 5.1. DPD Enzyme Assay in Peripheral Blood Mononuclear Cells (PBMCs)

While the liver is the main site of DPD activity, it cannot be non-invasively sampled to screen for DPD deficiency. Peripheral blood mononuclear cells (PBMCs) express functional DPD, and modest correlation has been noted between DPD activity measured in PBMCs and liver biopsies from the same patients [68,69], making them an attractive minimally invasive proxy for liver DPD function. Additionally, PBMCs are easily fractionated from whole blood using Ficoll-Paque [69], and PBMC DPD activity has been used to identify and characterize multiple deleterious *DPYD* variants (e.g., [70,71]). To measure DPD activity, PBMC lysates are incubated with labeled 5-FU, and degradation products are separated and measured using HPLC and a radio-isotope detector or mass spectrometry, depending on the type of label used [63,72].

Despite well-established methods for measuring DPD activity in PBMCs, the technical and time-consuming nature of the assay limits its use to primarily the research setting. In addition, numerous contributors to variability within the assay have been identified. For example, the choice of anticoagulant and time between blood collection and PBMC isolation can lead to variable capture of cellular types and inconsistent measurements of DPD activity [73,74,75]. The activity of DPD measured in PBMCs also displays a circadian rhythm with as much as a two-fold variation in a 24 h period [63,76], meaning that the timing of blood collection should ideally be standardized. The number of freeze–thaw cycles that PBMCs or lysates undergo has also been shown to greatly impact the measured DPD activity [69]. Because of these technical and practical limitations, PBMC DPD activity is not routinely used to screen patients for DPD deficiency, and to our knowledge, no commercial laboratories currently offer this test.

### 5.2. Pretreatment Uracil or Dihydrouracil:Uracil Ratio

As an alternative approach to estimating systemic DPD function, the levels of uracil (U) and the DPD-metabolism product dihydrouracil (UH_2_) can be measured in blood plasma [77,78]. If an individual is DPD-deficient, the catabolism of U to UH_2_ is reduced, resulting in elevated U and a reduced ratio of UH_2_:U. Exceptionally high levels of plasma U have been shown to indicate complete DPD deficiency and be predictive of elevated risk for severe 5-FU toxicity [79,80,81]. Threshold levels of plasma U have been proposed as indicative of DPD deficiency [79,80,81]. However, these cutoff levels have not been clinically validated as predictive of severe toxicity [67], and no prospective clinical trials have demonstrated that pre-treatment metabolite levels or ratios can be used to improve patient safety. In addition, extreme center-to-center differences have been reported for metabolite measures [38,66,67], and both circadian variation and food intake have been shown to affect plasma metabolite levels [76,82]. While the high variability inherent in these assays limits the interpretation of results at the individual level, the method has been highly useful as a research tool to identify DNA biomarkers of DPD deficiency.

### 5.3. 2-^13^C-Uracil Breath Test

The 2-^13^C-uracil breath test was developed as a modification of the 2-^13^C-urea breath test that was used to screen for *Helicobacter pylori* infection [83]. For this test, subjects ingest an aqueous solution of 2-^13^C-uracil. The levels of ^13^CO_2_ are subsequently measured in exhaled breath at various time intervals using IR spectroscopy. An initial study demonstrated that the amount of ^13^CO_2_ released and detected by the infrared detector was proportional to the level of DPD activity present [83]. A later study showed that the method had only a moderate ability to identify patients who would later experience severe 5-FU toxicity [84]. While this test has been shown to be non-invasive and rapid, the need for a specialized UBiT-IR300 spectrophotometer and the high cost of 2-^13^C-uracil likely contributed to the lack of further development and the adoption of this method for detecting DPD deficiency.

### 5.4. Oral Uracil Loading Test

The oral uracil loading test combines components of both the 2-^13^C-uracil breath test and the plasma UH_2_:U test. Like the breath test, a standardized test dose of uracil is administered to the subject as an aqueous solution. The ratio of UH_2_ to U is then measured in blood plasma at a set time. The rationale is that the catabolism of the bolus of uracil will correlate with systemic DPD function. The high uracil dose that is used is expected to surpass homeostatic levels of steady-state U and UH_2_, which can be affected by other pathways beyond DPD, thereby potentially offering a better indication of DPD function. Using PBMC DPD activity as a standard, the test showed promising sensitivity and specificity for identifying patients with reduced DPD activity [85,86]. Because the test uses unlabeled U, it is cost-effective to administer compared to the solution of labeled 2-^13^C-uracil needed for the breath test. However, the test does require an additional prolonged office visit to accommodate administration of the test dose and collection of a plasma specimen 2 h later [86]. Additionally, the performance of this test at predicting 5-FU toxicity has not been determined, nor have actionable levels for 5-FU dose adjustment been defined based on loading test results. As such, this test is also not widely used.

### 5.5. Therapeutic Drug Monitoring

Therapeutic drug monitoring (TDM) is another approach for identifying DPD deficiency. Early TDM research utilized subtherapeutic test doses of 5-FU that were administered to patients, and the circulating levels of 5-FU and metabolites were directly measured in blood thereafter using a variety of analytical approaches (e.g., [87,88,89,90]). With this method, there is concern that the test dose of 5-FU, although low, could still elicit adverse toxicity in severely DPD-deficient patients. Furthermore, because 5-FU is administered for diagnostic, not therapeutic, purposes, additional concerns were raised regarding potential impacts on tumor therapeutic resistance. Additional research has focused on applying TDM during therapy as a means of optimizing the dose of 5-FU with the goal of maintaining metabolite levels within the therapeutic range [91,92]. The most recent genotype-guided dose adjustment guidelines for 5-FU published by CPIC also recommend that TDM be used in patients who receive a reduced dose of 5-FU due to carrier status for a deleterious *DPYD* variant to optimize the dose to remain within the therapeutic range [20].

## 6. Current Regulatory Status of DPD Testing

Recommendations for pre-treatment testing for DPD deficiency vary by region/country, with the most prominent guidance at present coming from the European Medicines Agency (EMA). In 2020, the EMA published a direct healthcare professional communication (DHPC) that recommends testing for DPD deficiency prior to 5-FU treatment [93]. Additional regions within Europe have published their own, more specific, guidelines for testing, including a consortium of clinicians and researchers from Germany, Switzerland, and Austria [94], the National Health Service (NHS) of the United Kingdom [95], the Netherlands [96], and France [97]. U.S. medical organizations, including the Food and Drug Administration (FDA), National Comprehensive Cancer Network (NCCN), and the American Society for Clinical Oncology (ASCO), have not yet provided recommendations for universal pretreatment genotyping. Even though specific testing recommendations have not been given, the FDA does list “intermediate/poor metabolizer *DPYD* genotypes” as risk factors for severe or lethal toxicity on the “FDA table of pharmacogenetic associations with data supporting therapeutic management” [98], and the NCCN notes strong links between *DPYD* variants and toxicity risk as well as the potential benefits of testing [99].

## 7. Economic Considerations for *DPYD* Testing

Studies into the cost-effectiveness of pre-treatment testing for DPD deficiency have, to date, been limited to those that have used genotyping; the cost-effectiveness of phenotypic tests is unknown at present. Two studies in the Netherlands that used prospective genotyping prior to 5-FU treatment demonstrated that upfront genotyping for *DPYD* variants was modestly cost-saving, with the degree of cost-effectiveness most sensitive to hospitalization risk in variant carriers, the number and frequency of genotypes tested, and the cost of the test itself [100,101]. In a retrospective analysis of 20 colorectal cancer patients who developed severe neutropenia, Spanish researchers concluded that *DPYD* would be cost-effective if at least 2.1 cases of neutropenia were avoided out of 1000 patients tested [102]. A study of 134 patients that received first-line fluoropyrimidine therapy for colon cancer in Ireland similarly concluded that pre-treatment *DPYD* testing could be cost-saving, using data from [103]. A study of 550 colorectal cancer patients in Italy who were treated with fluoropyrimidines and retrospectively genotyped concluded that patients with deleterious *DPYD* variants incurred higher costs associated with managing toxicity than non-carriers and were at elevated risk for hospitalization related to toxicity [104]. Another retrospective study conducted in Italy showed that carriers of deleterious *DPYD* variants had higher healthcare-associated costs, poorer survival, and lower quality of life metrics [105]. Overall, these data indicate that the use of genetic testing to identify DPD-deficient patients is likely cost-saving to the healthcare industry and patients as a whole.

## 8. Conclusions

Deficiency of DPD is strongly linked to an increased risk of severe and potentially fatal toxicity to the commonly used chemotherapeutic 5-FU. Many methods have been used to identify patients with DPD deficiency. While phenotype-based tests have been instrumental in the research setting, genetic tests currently show the greatest promise for 5-FU dose individualization due to well-defined risk and dose-adjustment metrics for variant carriers. Recommendations for testing have been gaining momentum, with the EMA publication of guidelines for universal DPD testing prior to 5-FU use likely serving as what will be viewed in hindsight as a pivotal policy implementation in the field.

## Figures and Tables

**Figure 1 cancers-14-03207-f001:**
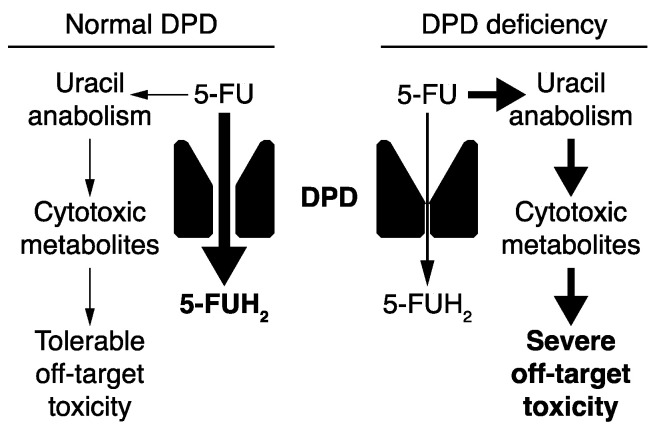
Overview of 5-FU metabolism showing that the catabolic pathway is the dominant pathway unless DPD deficiency causes a shift in 5-FU metabolism toward anabolism, resulting in increased risk for severe treatment-related toxicity.

**Table 1 cancers-14-03207-t001:** Reference information for commonly tested *DPYD* variants associated with DPD deficiency and increased risk of severe 5-FU toxicity.

rsID	RefSeqGene ID (LRG_722 NG_008807.2)	Transcript Change (NM_000110.3)	Amino Acid Change (NP_000101.2)	Other Names	Functional Impact
rs3918290	g.476002G>A	c.1905+1G>A	N/A ^1^	IVS14+1G>A, *2A	Completely deleterious
rs55886062	g.410273T>G	c.1679T>G	p.I560S	*13	Severely deleterious
rs67376798	g.843669A>T	c.2846A>T	p.D949V	-	Partially deleterious
rs75017182	g.346167C>G	c.1129-5923C>G	N/A ^2^	HapB3, rs56038477 (c.1236G>A, p.E412E) ^3^	Partially deleterious
rs115232898	g.226586A>G	c.557A>G	p.Y186C	-	Partially deleterious

N/A, not applicable. ^1^ Does not directly encode for an amino acid change but causes alternative splicing and the in-frame deletion of exon 14. ^2^ Does not directly encode for an amino acid change; causes non-obligate alternative splicing that introduces a frameshift and premature stop codon. ^3^ The rs56038477 variant is in strong LD with the causal variant (rs75017182), can be assessed using exome-level data, and is often used as a proxy for rs75017182.

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
