# Peer review of "Testing for Dihydropyrimidine Dehydrogenase Deficiency to Individualize 5-Fluorouracil Therapy"

_cancers, 2022, doi:10.3390/cancers14133207_

Round 1

Reviewer 1 Report

Congratulations for your hard work!

The article regarding Dihydropyrimidine Dehydrogenase Deficiency to 2 Individualize 5-Fluorouracil Therapy is an interesting one not only for clinical purposes, but also for clinical research. The topic is original and adds avalue to the subject area.

The paper is very well written, with a strong and clear data and statistical analysis.The conclusions are consistent with the evidence and arguments presented. 

Author Response

Thank you for reviewing the manuscript and the positive comments. We have performed a final check of the manuscript for grammar and spelling, making changes as noted.

Reviewer 2 Report

The review reports a comprehensive overview of the methods available to detect response in toxicity to c fluorouracil treatment.
The review covers the topic in all its aspects is clearly written and includes a large number of up-to-date literature references to support the topics discussed.

Author Response

(The authors gave the same response as above.)

Reviewer 3 Report

This article is written so that it can be quickly followed. It is also correctly structured. To enrich some of the points discussed in this article, I mention some aspects the authors could consider below.

Sometimes the toxicities graded by WHO are considered to exclude the 5-FU treatment. For example, 188 gastrointestinal cancer patients were given a test dose of 5-FU at 250 mg/m2 2 weeks before starting the planned 5-FU treatment of 370 mg/m2 plus L-folinic acid at 100 mg/m2 for 5 days every 4 weeks. Drug levels were examined by high-performance liquid chromatography (HPLC), and toxicities were graded according to WHO criteria. Of 188 patients, 3 (1.6%) had marked alterations of 5-FU/5-FDHU pharmacokinetics, and they were excluded from 5-FU treatments and treated with irinotecan, which was well tolerated. Unfortunately, this method has not been validated in a controlled fashion in the literature (Bocci et al., 2006).

The NIH Genetic Testing Registry, GTR, displays genetic tests for the DPYD gene, TYMS gene and the capecitabine drug response. The DPYD*2A variant is the most commonly tested. However, a negative result does not mean the individual does not have DPD deficiency. Clinicians should refer to the specific testing laboratory for complete information on the test. CPIC provides a table of minor allele frequencies for DPYD variants per ethnic population, which may be helpful when determining what type of test or panel is most informative for any individual. The GTR provides a list of biochemical tests that assess the levels of thymine and uracil analytes and the activity of the enzyme DPD.

Although the article is primarily oriented to DPD deficiency in treating cancer patients, it would be good to mention that in patients with a complete DPD deficiency (MIM 274270), a considerable variation in the clinical presentation has been observed, ranging from severely (neurologically) affected to symptomless 

Author Response

This article is written so that it can be quickly followed. It is also correctly structured.

Thank you for the positive comments.

Sometimes the toxicities graded by WHO are considered to exclude the 5-FU treatment. For example, 188 gastrointestinal cancer patients were given a test dose of 5-FU at 250 mg/m2 2 weeks before starting the planned 5-FU treatment of 370 mg/m2 plus L-folinic acid at 100 mg/m2 for 5 days every 4 weeks. Drug levels were examined by high-performance liquid chromatography (HPLC), and toxicities were graded according to WHO criteria. Of 188 patients, 3 (1.6%) had marked alterations of 5-FU/5-FDHU pharmacokinetics, and they were excluded from 5-FU treatments and treated with irinotecan, which was well tolerated. Unfortunately, this method has not been validated in a controlled fashion in the literature (Bocci et al., 2006).

The reviewer highlights two important issues that are discussed in the review paper. First that toxicities, which can be graded by a variety of standards, including CTCAE or WHO criteria, can result in discontinuation of therapy for patient safety. Changes to the treatment plan in response to toxicity differ greatly and are likely impacted by several variables. These likely include the severity of the toxicity, the treatment regimen, whether different treatments exist for a given cancer type/stage, local treatment “norms,” past physician experience, etc. In the end, changes to the treatment regimen, and the decision to continue 5-FU, adjust the dose, or discontinue use, are left to the judgement of the treating clinician and the patient. Because this is a complicated issue without clear globally accepted guidance, and it outside of the main focus of the manuscript (how to test for DPD deficiency), we chose not to discuss the issue at length. We will note that the introduction does provide a summary of treatment-related toxicity statistics from large and well-controlled trials. A review of guidance available for dose adjustment when the cause of toxicity is known to be a genetic variant is also included in section 2.

The second important issue involves how therapeutic dose monitoring (TDM) can potentially be used to identify individuals with DPD deficiency. The study summarized by the reviewer used a 5-FU monotherapy prior to initiation of multi-drug therapy and monitored conversion of 5-FU to the DPD-catabolized metabolite as a method to identify patients with suspected DPD deficiency. This is very similar to the approach first published by Ciccolini et al., 2004 (ref 87). In 2005, two additional groups reported similar approaches to dose monitoring, Di Paolo et al., 2005 (ref 88), and Remaud et al., 2005 (ref 89). More recently, Beumer et al., 2009 (ref 90), performed a multi-center analysis and validation of n immunoassay to measure circulating 5-FU levels for TDM. These earliest studies, as well as the more recent multi-center study, are discussed in section 5.5 of the manuscript. We acknowledge that there are many additional publications related to TDM that we were unable to cite in the review. In the case of the referenced paper by Bocci et al, 2006, it is noted that the pharmacokinetic methods that were used are similar to those that were reported earlier by this group in Di Paolo et al., 2005 (ref 87), so we decided to cite the earlier publication from the group.

The NIH Genetic Testing Registry, GTR, displays genetic tests for the DPYD gene, TYMS gene and the capecitabine drug response. The DPYD*2A variant is the most commonly tested. However, a negative result does not mean the individual does not have DPD deficiency. Clinicians should refer to the specific testing laboratory for complete information on the test. CPIC provides a table of minor allele frequencies for DPYD variants per ethnic population, which may be helpful when determining what type of test or panel is most informative for any individual. The GTR provides a list of biochemical tests that assess the levels of thymine and uracil analytes and the activity of the enzyme DPD.

We thank the reviewer for summarizing these sources of pharmacogenetic information. In our experience the information presented on the GTR is not up to date, and we therefore chose to focus on the current status of tests without referencing specific labs and instead referencing curated and peer-reviewed guidelines (e.g., CPIC), as well as global/governmental guidelines where they exist. We direct the reviewer to our discussions of genetic tests and interpretations, including CPIC guidelines, in section 4 and to our discussion of biochemical (ie, phenotypic tests) in section 5 of this review paper. Regulatory considerations are discussed in section 6. The limitations associated with “race-based genetic tests,” also mentioned by the reviewer, are discussed in section 2, lines 111-123.

Although the article is primarily oriented to DPD deficiency in treating cancer patients, it would be good to mention that in patients with a complete DPD deficiency (MIM 274270), a considerable variation in the clinical presentation has been observed, ranging from severely (neurologically) affected to symptomless 

As indicated by the reviewer, this review is focused on the pharmacological syndrome of DPD deficiency in the context of cancer therapy, and the manuscript is under consideration for publication in Cancers. There is very little that is actually known about the pediatric DPD deficiency mentioned by the reviewer, and, while the condition has been assigned a MIM #, it remains unclear if the condition is truly due to DPD deficiency or if some subjects with a range of neurological, developmental, and metabolic disorders also happen to carry variants that cause DPD deficiency. To date, mechanistic studies have not provided solid evidence directly linking the physiological presentation in these children to DPD deficiency, and attempts to ameliorate the condition by administering DPD metabolites have been unsuccessful. These findings further support the hypothesis that DPD deficiency itself might not be the underlying cause, or at least the sole cause. The reviewer is also correct that there is variable presentation of the disease in individuals with complete DPD deficiency (typically homozygous carriers of the *2A variant). Adding further questions to the link, similar conditions have been observed in children with partial DPD deficiency and also in children that have normal DPD function. Because of the lack of clear clinical link between the condition and DPD deficiency, not being relevant to cancer, and in the interest of space, we feel that discussion of this controversial and tangential topic would distract from the topic of this review.

Reviewer 4 Report

This paper has clearly discussed the effect of DPD deficiency on 5-Fluorouracil therapy. Diasio and Offer put an effort into demonstrating the methods to identify DPD deficiency in patient samples and the limitations which are needed to be improved in future work. This review is well written and qualified to be published on cancers. 

Author Response

Thank you for reviewing the manuscript, and we appreciate the positive comments.